# Planning with Spatial and Temporal Abstraction from Point Clouds for Deformable Object Manipulation

**Xingyu Lin** ∗†        **Carl Qi** ∗†        **Yunchu Zhang** †        **Zhiao Huang** ‡

**Katerina Fragkiadaki** †        **Yunzhu Li** §        **Chuang Gan** ¶        **David Held** †

**Abstract:** Effective planning of long-horizon deformable object manipulation requires suitable abstractions at both the spatial and temporal levels. Previous methods typically either focus on short-horizon tasks or make the strong assumption that full-state information is available. However, full states of deformable objects are often unavailable. In this paper, we propose PlAnning with Spatial and Temporal Abstraction (PASTA), which incorporates both spatial abstraction (reasoning about objects and their relations to each other) and temporal abstraction (reasoning over skills instead of low-level actions). Our framework maps high-dimension 3D point clouds into a set of latent vectors and plans skill sequences with the latent set representation. Our method can solve challenging, novel sequential deformable object manipulation tasks in the real world, which require combining multiple tool-use skills such as cutting with a knife, pushing with a pusher, and spreading dough with a roller. Additional materials can be found on our project website.[1]

**Keywords:** Long-horizon Planning, Deformable Object Manipulation

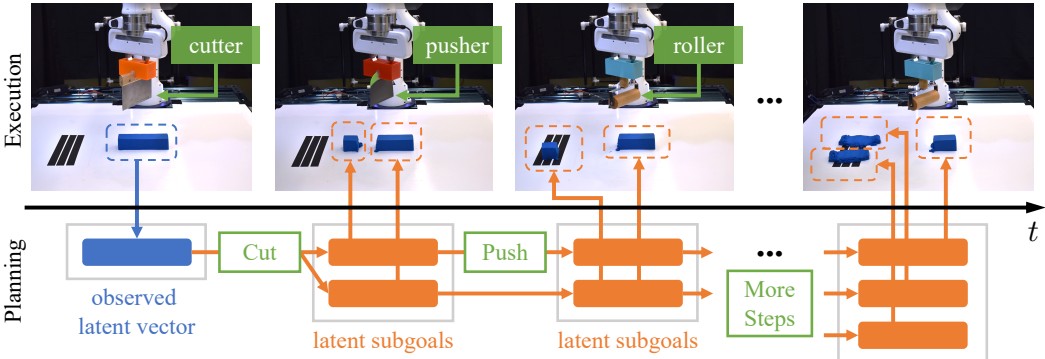

**Figure 1: Long-horizon dough manipulation with diverse tools.** Our framework is able to solve long-horizon, multi-tool, deformable object manipulation tasks that the agent has not seen during training. The illustrated task here is to cut a piece of dough into two with a cutter, transport the pieces to the spreading area on the left (with a high-friction surface) using a pusher, and then flatten both pieces with a roller.

## 1 Introduction

Consider a typical cooking task of making dumplings from dough. People plan over which piece of dough to manipulate and which tool to use in sequence, incorporating both spatial and temporal abstractions. A spatial abstraction reasons about objects, parts, and their relations to each other, such as reasoning about pieces of dough instead of reasoning about individual dough particles; such a

---

∗ equal contribution; † CMU ‡ UC San Diego; § Stanford University; ¶ UMass Amherst & MIT-IBM Lab
[1] https://sites.google.com/view/pasta-plan

6th Conference on Robot Learning (CoRL 2022), Auckland, New Zealand.

spatial abstraction enables efficient planning and compositional generalization. On the other hand, a temporal abstraction incorporates abstract actions represented as a set of skills. With abstract actions, the agent plans for the types and parameters of the skills to execute over a period of time, instead of making plans for low-level actions such as joint torques at each time step. Temporal abstractions allow planning at the skill level, enabling more efficient optimization for solving long-horizon tasks. An autonomous robot that operates in unstructured environments should be able to reason about world dynamics using high-level spatial and temporal abstractions instead of reasoning only over the physical state, raw sensory observation, or low-level robot actions.

The research community is making rapid progress towards developing state abstractions for manipulating deformable objects, including key points [1, 2], graphs [3, 4], dense object descriptors [5], or implicit functions [6, 7]. However, most of these approaches do not make abstractions at the temporal level, limiting their use to short-horizon tasks. Methods are also being developed with temporal abstractions, planning over a set of skills to solve long-horizon tasks [8, 9, 10]. However, the lack of spatial abstraction severely limits their generalization ability. Therefore, it remains a key question in robot learning on how to learn spatial and temporal abstractions within a unified framework for complex and long-horizon manipulation tasks.

In this work, we focus on the challenging task of sequential deformable object manipulation, as shown in Figure 1. We consider a set of dough manipulation tasks that require sequentially applying different skills using multiple tools to manipulate dough, such as spreading using a roller, cutting using a knife and pushing using a pusher, where the longest task requires applying 6 skills in sequence. Deformable objects like dough have nearly infinite degrees of freedom. As such, in this work, we dynamically cluster points in a point cloud into different groups and learn a point cloud encoder to map each element in the group into a latent vector. In this way, we obtain a compositional 3D set representation of the state space. Given an observation and a target point cloud, we then sample skill sequences with subgoals generated in this latent space. We learn skill abstraction modules to determine the feasibility and score of each skill sequence and use them for planning.

Our contribution of this paper is a framework that PlAns with Spatial and Temporal Abstraction (PASTA) by learning a set of skill abstraction modules over a 3D set representation. Our framework can compose a set of skills to solve complex tasks with more entities and longer-horizon than what was seen during training. We show that PASTA significantly outperforms an ablation that performs planning with a flat representation without a spatial abstraction (e.g. without a set representation). Finally, our planner can be trained in simulation and transferred to the real world.

## 2    Related Work

**Model-based Planning for Sequential Manipulation.** One line of research for sequential manipulation is Task and Motion Planning (TAMP). TAMP systems typically assume known object states and known effects for the action operators [11, 12, 13, 14, 15]. However, it is difficult to estimate states and dynamics for unknown objects or from partial observations. While recent works have made progress in learning certain components of the system, such as the logical states [16] from high dimensional observations or learning action models [17, 18, 19] from interactions, they still require either known states or known action operators. In contrast, we do not assume known states or action operators, and learn a 3D set representation as well as the action model with the representation.

Another approach learns dynamics directly from visual observations [3, 20, 21]. Most of these works focus on learning a one-step dynamics model for planning short-horizon tasks. A few works learn the dynamics model over a set of skills and use it for sequential manipulation of rigid objects [22, 23] or deformable objects [8]. However, these works do not use an object-centric representation and thus cannot easily generalize to more complex scenes. In contrast, our framework unifies both temporal and spatial abstraction and can perform long-horizon manipulation for complex tasks with more objects than in previous work, as we will show.

**Planning with Spatial Abstraction.** Prior works leverage spatial abstraction to facilitate solving tasks that involve complex dynamics and high-dimensional observations. These works either model a

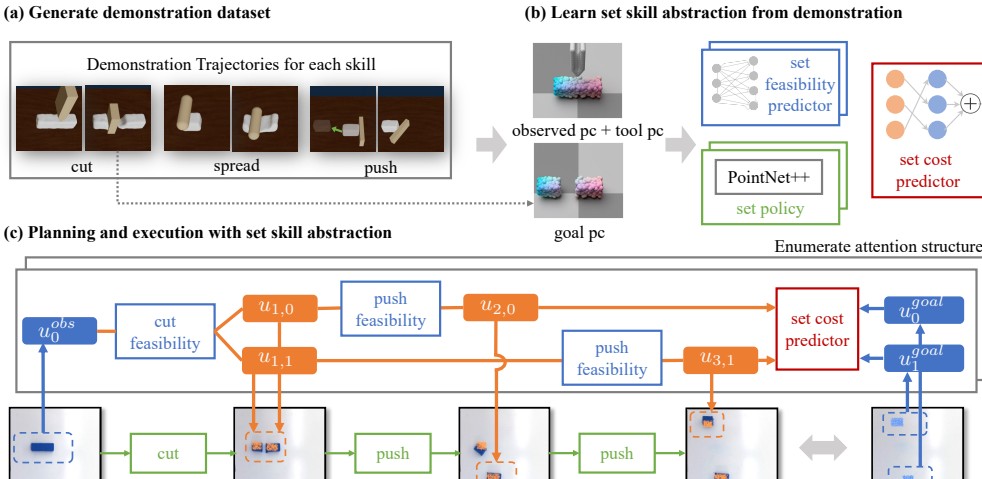

**Figure 2: Overview of our proposed framework PASTA.** (a) We first generate demonstration trajectories for each skill in a differentiable simulator using different tools. (b) We then sample point clouds (pc) from the demonstration trajectories to train our set skill abstraction modules. (c) We map point clouds into a latent set representation and plan over tool-use skills to perform long-horizon deformable object manipulation tasks. $P^{obs}$, $P^{goal}$ are the observation and target pc; $u_{i,j}$ denotes component $j$ at step $i$. The example shows our method performs the CutRearrange task, which requires cutting the dough into two pieces with a knife and transporting each piece to its target location.

compositional system with Graph Neural Networks (GNN) [3, 4, 7, 24, 25] or learn policies directly from object-centric representations [26, 27]. These works demonstrate compositional generalization, but they learn policies or one-step dynamics models for planning, which can be difficult for solving long-horizon tasks. In contrast, our framework connects temporally extended spatial abstractions with a feasibility predictor to plan over a longer time horizon. Xu et al. [28] learns a planner grounded on object-centric visual observation and can solve long-horizon tasks. However, its planner takes a symbolic goal and plans in a predefined symbolic domain to output symbolic subgoals. In contrast, our method does not require defining a symbolic planning domain.

**Deformable Object Manipulation.** Deformable objects have nearly infinite degrees of freedom and complex dynamics, making them very challenging to manipulate. Previous works have explored pouring liquid [6, 29, 30], rope manipulation [31, 32], and cloth manipulation [33, 34, 4, 35, 36]. Other papers have also explored manipulating elasto-plastic objects such as deforming them by grasping [3, 37], rolling [38, 39, 40], or cutting [41]. However, these works mostly only consider manipulation with one skill at a time. In contrast, we consider the task of sequential manipulation using multiple tools. The one exception is DiffSkill [8], where multiple skills are chained together. However, DiffSkill uses RGB-D images to represent the scene. In contrast, we use a 3D set representation that separately encodes each entity in the scene, enabling compositional generalization to tasks with more objects and longer-horizon. Furthermore, we use point clouds as the input and we are able to transfer our planner from simulation to the real world.

## 3 Method

Given a point cloud of the dough $P^{obs}$ and a goal point cloud $P^{goal}$, our objective is to execute a sequence of actions $a_1, ..., a_{T_{tot}}$ that minimizes the distance between the final observed point cloud and the goal $D(P^{obs}_{T_{tot}}, P^{goal})$ where $P^{obs}_{T_{tot}}$ is the observation point cloud at time $T_{tot}$. We aim to solve long-horizon tasks that require chaining multiple skills in novel scenes with more objects than training. To do so, we present a general framework that incorporates spatial and temporal abstractions, as summarized in Figure 2. We use point clouds as input to all our modules to enable easier transfer from simulation to the real world and to enable robustness to changes in viewpoint.

We assume access to an offline dataset of demonstration trajectories $\mathcal{D}_{demo}$ from $K$ skills, where each trajectory demonstrates one of the skills using one tool. We can learn skill policies by imitation learning from these demonstrations. To chain these skills to solve long-horizon tasks, we train a set of skill abstraction modules (Sec. 3.3) which can be used for planning in the latent space (Sec. 3.1).

## 3.1 Spatial Abstraction from Point Clouds

**Scene Decomposition:** First, we describe our spatial abstraction of the point cloud observation. Given a point cloud $P \in \mathbb{R}^{N \times 3}$, we first cluster the points into different components based on their proximity in space. In this paper, we apply DBSCAN [42] to $P$ and group points into a set of entity point clouds $\{P_i \in \mathbb{R}^{N_i \times 3}\}_{i=1...C}$ by separating points from high-density regions into different clusters. While other works on scene decomposition can also be used [43], we find this simple method to be sufficient for our tasks.

**Entity Encoding:** Planning directly in the high-dimensional space of point clouds is inefficient. To enable efficient planning in a latent space, we train a point cloud variational autoencoder (VAE) [44]. The VAE model includes three modules: A point cloud encoder $\phi : \mathbb{R}^{N_i \times 3} \to \mathcal{U}$ that maps each entity point cloud to a latent vector, a decoder $\psi : \mathcal{U} \to \mathbb{R}^{N_i \times 3}$ that maps from a latent space back into a point cloud, and a prior distribution over the latent space $p_u : \mathcal{U} \to [0, 1]$ which can be used to generate samples from the latent space during planning. We can then encode the point clouds $\{P_i\}$ into a set of latent vectors: $\{u_i\}_{i=1...C}$ as our set latent representation. We further achieve translation invariance by separating the translation from its shape embedding. See the Appendix for details.

## 3.2 Learning Skills by Imitation

Given the demonstration trajectories $\mathcal{D}_{demo}$ of $K$ skills, we first distill these trajectories into $K$ closed-loop policies. The input to the policy for the $k^{th}$ skill $\pi_k$ is a subset of the observed point clouds $\{P_i^o\}$ and goal point clouds $\{P_j^g\}$, and a tool point cloud $P_k^{tool}$. The policy only sees a subset of the dough point cloud and goal point cloud in the scene to enable compositional generalization to scenes with more objects. For example, a dough-spreading policy will only see the dough being spread. To achieve this, we train each set point cloud policy with behavior cloning and hindsight relabeling [45] on the demonstration dataset with an attention mask that filters out the non-relevant entity point clouds. During planning, this attention mask will be provided by the planner. The policy outputs an action at each timestep to control the tool directly.

## 3.3 Neural Spatial and Temporal Abstraction

We assume that each skill learned from the demonstration is only capable of performing a single-stage task with a single tool for a single object. To solve longer-horizon tasks, we further learn a feasibility predictor and a cost predictor. They can be used to plan subgoals that chain the skills into a sequence, such that each subgoal is feasible for the corresponding policy to reach and the final subgoal reaches a given goal. Additionally, all these modules take in the set representation $\{P_1 \ldots P_C\}$ as input to achieve compositional generalization.

**Set Feasibility Predictor** Similar to DiffSkill [8], we train a feasibility predictor $f_k(U^o, U^g)$ for each skill, where $U^o = \{u_i^o\}_{i=1...N_o}, U^g = \{u_j^g\}_{j=1...N_g}$ are latent set representations of an observation and goal point cloud respectively. The feasibility predictor outputs a value in $[0, 1]$ denoting if the goal can be reached from the observation by executing the $k^{th}$ skill. In DiffSkill, the feasibility predictor uses a flat representation that takes in a single latent vector for all objects in the scene as input. However, as our skills such as cutting or spreading only need to take in a subset of the objects as input, we use the same attention method for the feasibility and assume that the feasibility predictor only takes as input a subset of the full set representation $\hat{U}^o \subseteq U^o, \hat{U}^g \subseteq U^g$, where $\hat{U}^o = \{u_i^o\}_{i=1...N_k}, \hat{U}^g = \{u_j^g\}_{j=1...M_k}$, Here, $N_k$ and $M_k$ are the number of components in the observation and goal for skill $k$. The number of components in the observation and goal can be different since the number of components can change before and after executing a skill; for example, the cut skill takes one component as observation and cuts it into two components. As another example

for robot assembly [46], the number of entities increases when a piece is disassembled into parts and the number of entities decreases when the parts are assembled. In this work, we manually define the number of entities $N_k$, $M_k$ per skill. Determining which subset to attend to when executing each skill can be difficult; we make this decision during planning and defer the details to Sec. 3.4. We parameterize $f_k$ to be invariant to permutation using max-pooling layers. See Appendix for details.

We train the feasibility predictor of skill $k$ with positive examples $\hat{U}^o$, $\hat{U}^g$, where the goal $\hat{U}^g$ can be reached from the observation $\hat{U}^o$ within $T$ timesteps by executing skill $k$. The negative examples are goal $\hat{U}^g$ that cannot be reached from the observation $\hat{U}^o$ using skill $k$. During training, we obtain positive pairs for the feasibility predictor by sampling two point clouds $(P^{obs}, P^{goal})$ from the same trajectory in the demonstration set. To find $\hat{U}^o, \hat{U}^g$, we first cluster the observation and goal point clouds into two sets $\{P_i^o\}, \{P_j^g\}$ respectively. Then, we match point clouds in the observation set to those in the goal set by finding the pairs of point clouds that are within a Chamfer distance of $\epsilon$: $\{(P_i^o, P_j^g) \mid D_{Chamfer}(P_i^o, P_j^g) < \epsilon\}$. We then remove these point clouds from the corresponding set, since these are the point clouds that have already been moved to the target location in the goal. We can then encode the remaining point clouds into $\hat{U}^o, \hat{U}^g$ as explained above. We generate hard negative samples by replacing one entity in the positive examples with a random latent vector.

**Set Cost Predictor** As we do planning in a latent space, we train a set cost predictor as our planning objective which determines how close a plan is to a given goal. The set cost predictor $C$ takes two latent set representation as input $U^o, U^g$. Since our tasks focus on matching each entity in the observation with one in the goal, we assume they have the same number of components, i.e. $|U^o| = |U^g| = N_c$. To compute the cost, we try to find the matching entity with the minimal matching cost: $C\left(\{u_i^o\}, \{u_j^g\}\right) = \arg\min_\sigma \sum_{i=1}^{N_c} c_\theta(u_i^o, u_{\sigma(i)}^g)$, where $\sigma$ is a permutation and $c_\theta$ is a cost prediction network parameterized by an MLP trained to predict the Chamfer Distance between the point clouds corresponding to the two latent vectors. This allows faster planning compared to first decoding latent vectors to point clouds and then computing their distance. Finally, optimization of the cost is done by performing Hungarian matching between the two sets containing latent vectors.

## 3.4 Planning with Set Representation

Given an observation and a goal point cloud $P^{obs}, P^{goal}$, we plan for the types of skills to apply in sequence, the attention for each skill (i.e. find $\hat{U}^o \subseteq U^o$), and the latent subgoals for each skill (i.e. the exact value for each latent vector in $\hat{U}^o$). As the simplest approach, we run a three-level nested optimization: In the top level, we exhaustively search over the combinations of skills to apply at each step, i.e. $k_1 \ldots, k_H$, where $k_h$ indexes the skill applied at the high-level step $h$. We only keep the sequences that end with the same set cardinality as the goal by ensuring that $\sum_{h=1}^{H} M_{k_h} - N_{k_h} = N_g - N_o$, where $M_{k_h}$ and $N_{k_h}$ are the number of observation and goal components for the skill applied at step $h$ and $N_o$ and $N_g$ are the number of components in the observed and target point clouds.

In the second-level optimization, we search over different attention structures. Denote the latent set at the high level step $h$ to be $U^h$. We formally define the attention structure at step $h$ to be $I^h$, which consists of a list of indices, each of length $N_{k_h}$, such that $I^h$ selects a subset from $U^{h-1}$ to be the input to the feasibility predictor, i.e. $\hat{U}^{h-1} = U_{I^h}^{h-1} \subseteq U^{h-1}$. Assume that we have $N_h$ components before applying skill $k_h$, i.e. $|U^{h-1}| = N_h$ and skill $k_h$ takes $K_h$ components as its observation. We can search over all $C_{N_h}^{K_h}$ combinations of attention structures. For components not considered by the skill, its latent vector will remain the same at step $h$. The combination of each skill attention yields an attention structure $\mathbf{I}$ for the whole plan, as illustrated in Fig. 2(c). For this level of optimization, we use a sampling-based procedure to avoid an exhaustive search over topologically equivalent attention structures. See the Appendix for how we do this efficiently.

In the low-level optimization, for each attention structure $\mathbf{I}$, we follow the optimization in DiffSkill [8]. We first sample multiple initializations for the set of latent subgoals $\mathbf{U}$, where each latent vector in the set is initialized from our generative model. We can then perform gradient descent to further optimize the latent subgoals on the following objective:

$$\underset{\mathbf{k,I,U}}{\arg\min} J(\mathbf{k,I,U}) = \prod_{h=1}^{H} f_{k_h}(\hat{U}^{h-1}, \hat{U}^h) \exp(C(U^H, U^g)), \tag{1}$$

where $\mathbf{k}$ is the skill sequence, $\mathbf{I}$ is the attention structure of the plan, $\mathbf{U}$ is the set of all latent subgoals, $\hat{U}^h = \{u_i^h\}_{i=1...M_{k_h}}$ are the latent subgoals at step $h$, $\hat{U}^0 = \hat{U}^o \subseteq U^o$ is the attended observed set, and $U^g$ is the goal set. Finally, we can use our policy to execute our plan by following each subgoal. A summary of our method can be found in Algorithm 1.

---

**Algorithm 1:** Planning with Spatial and Temporal Abstraction (PASTA)

---

**Input** : Demonstration Dataset $D_{demo}$, skill horizon $T$, planning horizon $H$, modules for neural skill abstraction $\pi_k, f_k, r_\theta$, Point Cloud VAE with encoder $\phi$, decoder $\psi$, prior $p_u$

1 **for** *each valid skill sequence* $k_1, \ldots k_H$ **do**
2      **for** *each valid attention structure* $I^1, \ldots I^H$ **do**
3          Initialize different latent subgoals $\hat{U}^1, \ldots \hat{U}^H$ from $p_u$ ;
4          Optimize latent subgoals $\hat{U}^1, \ldots \hat{U}^H$ according to Sec. 3.4 to obtain cost $J(\mathbf{k,I,U})$ ;
5 Choose skill sequence $\mathbf{k}$, attention structure $\mathbf{I}$, and subgoals $\mathbf{U}$ that minimizes $J(\mathbf{k,I,U})$ ;
6 **for** $h \leftarrow 0$ **to** $H - 1$ **do**
7      Decode the subgoal from $\hat{U}^h$ using the decoder $\psi$ ;
8      Execute policy $\pi_{k_i}$ following the subgoal ;

---

## 4 Experiments

Our experiments are categorized into three parts: In Sec. 4.1, we describe the experimental setups and the baselines we consider. In Sec. 4.2 and Sec. 4.3, we show that PASTA outperforms the baselines and ablate different components in our framework. In Sec. 4.4, we demonstrate that PASTA can be effectively transferred to the real world without any fine-tuning.

### 4.1 Simulation Tasks and Baselines

**Environment setups** We consider several long-horizon dough manipulation tasks and divide them into two categories. First, we consider the three tasks from Diffskill [8]: LiftSpread, GatherTransport, and CutRearrange. These tasks require the agent to sequentially compose at most two skills to spread, cut or transport the dough. We further propose two new generalization tasks: CutRearrangeSpread (CRS) and CRS-Twice, where there are more entities during testing than during training. Similar to prior work [8], we specify the minimal planning horizon required for each task. Our approach also succeeds when we increase the horizon up to twice as long. See the Appendix, Sec 4.5 for details.

**Generalization tasks** The CRS task provides a number of demonstration trajectories performing one of the three skills: Cutting with a knife, pushing with a pusher, and spreading with a roller. The demonstration of each skill only shows a tool manipulating a single piece of dough. During testing, the agent needs to cut a dough into two, transport one piece to a spreading area and then spread it. Generalization to more entities is required as there will be two entities in the scene, whereas during training there was only one entity. Can we do even more generalization? In the CRS-Twice task, we use the same agent trained on the CRS dataset and ask it to cut two pieces of dough from a chunk, transport both of them to a spreading area, and spread them both. This is a 6-horizon task with up to 3 entities in the scene, much more complex than the skill demonstrations the agent is trained on. Due to the long-horizon nature of CRS-Twice, we specify the skill skeleton and use receding horizon planning for all the planning-based methods. See the Appendix for details.

**Baselines** We consider several baselines in simulation: First, a gradient-based trajectory optimizer with oracle information (Traj-Opt), which can solve single-stage tasks for deformable object manipulation as shown in prior works [47]. Second, a model-free RL with Soft Actor Critic [48] with RGB-D image input (SAC-Image). Third, a SAC agent that takes in the dough, target dough, and tool point clouds as input, the same as our method (SAC-Point). Fourth is DiffSkill, a model-based planning

method from Lin *et al.* [8], which takes RGB-D images as input and has no spatial abstraction. The last one is Flat 3D, which extends DiffSkill to use 3D point clouds as input, with a "flat" 3D representation that encodes the whole scene to a single latent vector without any spatial abstraction.

**Metric** We specify goals as 3D point clouds of different geometric shapes such as boxes and spheres at specific locations. We report the normalized decrease in the Earth Mover Distance (EMD) approximated by the Sinkhorn diverge [49] computed as $s(t) = \frac{s_0 - s_t}{s_0}$, where $s_0, s_t$ are the initial and current EMD. We additionally set a threshold for the score to determine the success of a trial.

## 4.2 Comparison with Baselines in Simulation

| Task (Horizon) | DiffSkill tasks | | | Generalization tasks | |
|---|---|---|---|---|---|
| Method | LiftSpread (2) | GatherMove (2) | CutRearrange (3) | CRS (3) | CRS-Twice (6) |
| Traj-Opt (Oracle) [47] | 0.818 / 40% | 0.403 / 0% | 0.511 / 20% | 0.312 / 0% | 0.227 / 0% |
| SAC-Image [48] | 0.797 / 0% | 0.567 / 20% | 0.103 / 0% | 0.562 / 0% | 0.365 / 0% |
| SAC-Point [48] | 0.796 / 0% | 0.603 / 40% | 0.147 / 0% | 0.573 / 0% | 0.353 / 0% |
| DiffSkill-Image [8] | **0.920 / 100%** | 0.683 / 60% | 0.249 / 20% | -0.505 / 0% | - |
| Flat 3D (Ours) | * | * | 0.797 / 60% | -0.712 / 0% | -0.108 / 0% |
| PASTA (Ours) | 0.904 / **100%** | **0.715 / 100%** | **0.837 / 80%** | **0.896 / 100%** | **0.604 / 40%** |

**Table 1:** Normalized improvement and success rate of all methods on two sets of tasks: tasks in DiffSkill and tasks that require generalization to more steps and entities. For CRS and CRS-Twice, training data only contains skills operating on **one** component of dough but at test time there are more than two components. Only the best-performing baselines in CRS are evaluated on CRS-Twice. For LiftSpread and GatherMove, we consider the whole scene as a single spatial abstraction, so Flat 3D is equivalent to PASTA.

Table 1 shows the quantitative results of simulation tasks. First, we show that using a 3D representation is beneficial to planning and complex manipulation, as PASTA matches DiffSkill in LiftSpread and outperforms it in all the other tasks. Second, we highlight PASTA's compositional generalization power in CRS and CRS-Twice, in which there are additional components of dough at test time. Effectively using spatial abstraction to model the scene, PASTA achieves 100% success rate in CRS and retains a good performance in CRS-Twice. All the baselines, especially the planning baselines (DiffSkill, Flat 3D) fail dramatically, as they can only produce plans that consist of scenes seen in training. Impressively, PASTA is the only approach that reaches a non-zero success rate on these tasks that require compositional generalization. PASTA has a computational complexity of $\mathcal{O}(K^H \cdot |\mathbf{I}| \cdot \mathbf{T_G})$, where $K$ is the number of skills, $H$ is the planning horizon, $|\mathbf{I}|$ is the number of attention structures and $T_G$ is the time it takes to solve each gradient-based optimization. We discuss more efficient planning algorithms in the limitation section as well as in the Appendix, Sec. 5.

## 4.3 Ablation analysis

Table 2 shows the quantitative performance of each ablation and PASTA in CutRearrange. First, we consider a variant of feasibility predictor's training, which removes the hard negative

| Ablation Method | Performance / Success |
|---|---|
| No Hard Negatives Feasibility | 0.740 / 40% |
| No Sampling Planning | -0.455 / 0% |
| No Gradient Planning | 0.329 / 0% |
| PASTA (Ours) | **0.837 / 80%** |

**Table 2:** Ablation results from CutRearrange.

samples and only uses random negative sampling (No Hard Negatives), which halves the success rate. Second, we consider two variants of the planner, one without gradient-descent (No Gradient Planning) and one without sampling (No Sampling Planning). The results show that both components are crucial to planning. We provide more ablations on the policy and the planner in the Appendix.

## 4.4 Real World Experiments

Figure 3 shows our real world setup. We use a Franka robot with an Azure Kinect camera capturing the RGB-D observation of the workspace. The robot is equipped with a tool station [50] that allows an automatic change of tools. For real world "dough", we use Kinect Sand as a proxy because of its stable physical property. We transfer the feasibility predictor and cost predictor of PASTA directly from simulation and define heuristic controllers for the skills. For evaluation, we first generate a desired target point cloud and then reset the dough to its initial shape and record its point cloud. We

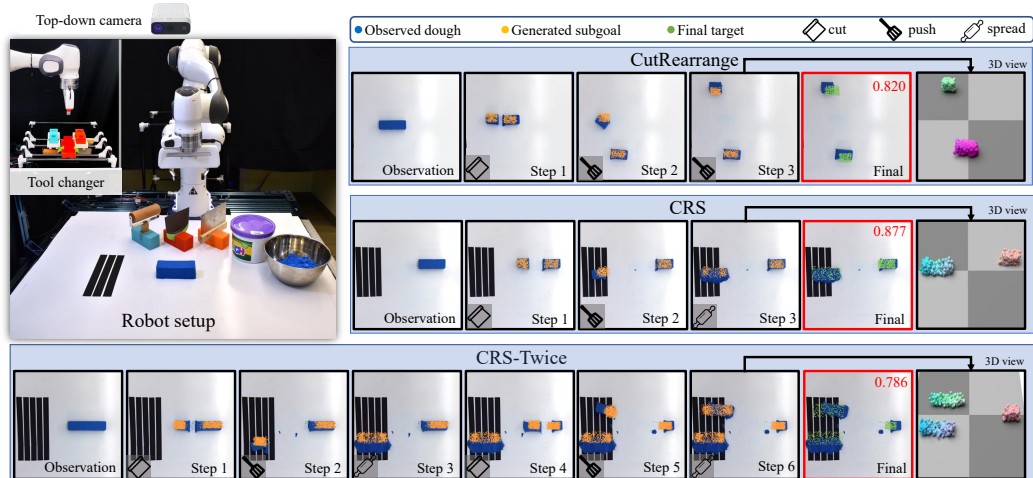

**Figure 3: Real world setup and execution with planned subgoals.** Our workspace consists of a Franka robot, a top-down camera, and a novel tool changer behind the robot that allows the robot to automatically switch tools. For each task, we show frames after executing a skill overlaid with the decoded point cloud subgoal; we report the final performance in red and overlay the ground truth target in green in the final frame. Additionally, we include a 3D view of the last generated subgoal to show the shape variations.

then plan using PASTA and then use the heuristic controllers to follow the plan. Finally, we report the normalized improvement EMD. We compare with the Flat3D method and also with human.

We evaluate three of the simulation tasks: CutRearrange, CRS, and CRS-Twice. For each task, we evaluate the same four initial and target shapes for all methods and report the performance in Table 3. Figure 3 shows the keyframes from the execution of PASTA. We overlay the planned subgoals as well as the final goal. PASTA performs on par with human in the real world, highlighting the robustness of our planner and the advantage of using 3D representation for sim2real transfer.

| Method | Task (Horizon) CutRearrange (3) | CRS (3) | CRS-Twice (6) |
|---|---|---|---|
| Flat 3D | $0.351 \pm 0.478$ | $0.007 \pm 0.429$ | - |
| PASTA (Ours) | $\mathbf{0.836 \pm 0.029}$ | $\mathbf{0.854 \pm 0.016}$ | $\mathbf{0.795 \pm 0.035}$ |
| Human | $0.910 \pm 0.014$ | $0.863 \pm 0.018$ | $0.895 \pm 0.013$ |

**Table 3:** Normalized improvement on real world tasks. Each entry shows the mean and std of the performance over 4 runs. Flat 3D does not produce any meaningful plan for CRS, so we do not evaluate it on CRS-Twice.

## 5 Conclusions and Limitations

In this work, we propose a planning framework named PASTA that incorporates both spatial and temporal abstraction by planning with a 3D latent set representation with attention structure. We demonstrate a manipulation system in the real world that uses PASTA to plan with multiple tool-use skills to solve the challenging deformable object manipulation tasks, and we show that it significantly outperforms a flat 3D representation, especially when generalizing to more complex tasks.

**Limitations:** First, we only transfer the planner to the real world and use heuristic controllers instead of the policy trained in simulation. This is due to the sim2real gap caused by the differences in dough's physical parameters, table friction, and occlusions from the robot arm. Prior work [38] shows promising results in transferring a closed-loop policy taking partial point clouds as input, and future work can explore better sim2real methods. Second, our planner exhaustively searches overall skill combinations and attention structures and does not scale well to longer sequences with more skills. Future work can incorporate more efficient search algorithms or priors to prune the search space. Finally, we rely on an unsupervised clustering method for entity decomposition and a point cloud VAE for mapping an observation to our latent set representation. Future work can incorporate self-supervised methods for learning the decomposition. For further discussion on the limitations and future work, please see Sec. 5 of the Appendix.

**Acknowledgments**

This material is based upon work supported by the National Science Foundation under Grant No. IIS-2046491, IIS-1849154, NSF award under AWD00001520, LG Electronics, Carnegie Mellon University's GSA/Provost GuSH Grant funding, and Amazon Research Award.

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
