# OpenReview forum: "Planning with Spatial-Temporal Abstraction from Point Clouds for Deformable Object Manipulation"
_robot-learning.org/CoRL/2022/Conference — CoRL 2022 Poster_

### Official Review · Reviewer_jQsA · 2022-07-29

**Originality:** Very Good
**Technical Quality:** Very Good
**Clarity Of Presentation:** Good
**Impact:** 4

**Recommendation:**

Weak Accept: I recommend accepting the paper, but will not argue for my recommendation if the majority of other reviewers have a different opinion.

**Summary:**

This work is about long-horizon elastoplastic object manipulation, including multiple tools. This paper emphasizes that their method incorporates spatial abstraction and temporal abstraction. Here, spatial abstraction is about decomposing a dense scene into a set of objects and representing them with embeddings, and temporal abstraction is about high-level reasoning over different skills. PASTA executes a trajectory by solving a two-step optimization problem where the first step finds an optimal sequence of skills and the second step finds an optimal sub-goal for each skill.


**Issues:**

- Doughs in real life are much more challenging to handle because it sticks to the desk and the tool. Also, dough's material property is time-variant. (Get firmer and less sticky as it dries). This seems to be a great huddle for PASTA's sim2real transfer. Including the authors' insight on integrating these uncertainties with an action generator will make this study much stronger.
- Justification of using tool point cloud: Why use the tool 'pointcloud' for Point Cloud Policy instead of the estimated pose and the type of the tool? For example, the tool can be illustrated with 6 numbers (x,y,z,roll,pitch,yaw) where the type of the tool can be expressed in one-hot [0 ... 1 0 ..]. I wonder how robust pointcloud-based tool input regarding the tool occlusion caused by the robot arm/gripper and dough.
- Why do you think decoupling subgoal and skill sequence for optimization is a great choice? Why not optimize them simultaneously?
Why is it better to use a reward 'predictor'? (Line 173) Can we convert the decode the latent codes and directly get the reward? For example, you can explicitly use grid search to get the similarity score.
- I feel that the author's heuristic controllers in the real world might conceal the sub-goal and action error. In detail, in 'Fig.3. CutRearrange' step 2 (top sand) shows that PASTA shows some error at 1) setting accurate subgoal and 2) moving the dough to the target place. Ironically, the errors devoted to increasing the 'Final' accuracies, which I think is odd. Can the authors explain this?

**Writing**
Explicitly making a section describing the training process and the entire loss formulation will help readers to follow the paper.
How much train and test data? What is their distribution?

**Quality Of The Limitations Section:**

Additional details required

**Reviewer Expertise:**

4: The reviewer is confident but not absolutely certain that the evaluation is correct

**Robotics Focus:**

Sufficient demonstration on hardware

**Strengths And Weaknesses:**

**Advantages**
- Suggest planning with tool changes and learning individual policies for each tool.
- Simulation and Real-world experiments using different types of elastoplastic material(sand and real dough).

**Weaknesses**
- One of the main limitations of this paper is that their ‘point cloud policy’ or the action generator is not evaluated in or transferred to real-world experiments. For the real experiment, they utilized heuristic controllers for the skills.
- PASTA does not allow feedback between each subgoal in a sequence (k_1, …, k_H). Planning in the long-horizon is a great direction; however, not updating subgoals during the long-horizon manipulation makes the method susceptible to disturbances.
Requires to specify the number of steps during tests. Performance efficiency and accuracy depending on user input can also be the limitation of this work.
- PASTA relies on manual pre-designed skill definitions (M_k_h, N_k_h).
- Lack of clarifications/typos in some texts
    - Line 86-88: I am not sure how ‘using 3D set representation extracted from point clouds’ can directly be linked to generalizing compositionally to tasks with "more objects and longer horizon”.
    - Line 130 : can you clarify the length of action from \pi? The output actions per each skill should be greater than 1, especially for spreading, which makes the robot move the roller back and forth.
    - Line 138 : definition of N_o and N_g
    -Typo in line 116: “We can the encode~ “

**Summary Of Recommendation:**

Elasto-plastic objects are generally hard to manipulate with robotic fingers or grippers, thus requiring tool use. This study is powerful in that it allows deformable object manipulation with multi-tools. However, I am hesitant to recommend this paper unless the authors address the major limitations. As discussed in 'issues', the major concerns are 1) partial sim2real transfer and potentially misleading evaluations and 2) lack of justifications in model choice/design (2-4 bullet points in issues). Other than that, thorough real-world analysis and experiment are great, and the paper overall was enjoyable to read.

---

> ### Author Response · Authors · 2022-08-25
> **Author response (Part 1 of 2)**
>
> Thank you for your detailed and insightful review. We have uploaded a new manuscript fixing all issues on presentation and we mark the differences in magenta. We have responded to each of your concerns below:
>
> > Concern 1: Point cloud policy is not transferred to the real-world
>
> First, our contribution is on planning and not on learning policies. Specifically, our contribution in this paper is a planning framework that incorporates a spatial and temporal abstraction. Long-horizon planning for manipulating deformable objects with multiple skills is a challenging task and has not been demonstrated in previous literature, neither in simulation nor in the real world. Our planner is also transferred to the real world to show that our planning method can work on real observations.
>
> We agree with the reviewer that transferring the policy can also be important, though separate from the challenge that we focus on. We have added more discussion about transferring the policy in the limitation section in our updated manuscript. The two difficulties of transferring the policy from simulation to the real-world are described below:
>
> * The different dynamics between simulation and the real world, including the physical properties of the dough and the table friction.
> * There are severe occlusion of the dough in real world due to the robot arm and the tool, making it more difficult to transfer a closed-loop policy.
>
> Since the focus of the paper is on planning, we leave these for future work.
>
> > Concern 2: Dough in real life is hard to manipulate. Including the authors' insight on integrating these uncertainties with an action generator will make this study much stronger.
>
> This is a good point.  We have updated the limitations section to discuss the sim2real transfer in more detail.  One possible approach is to train with domain randomization to make the policy more robust to changing dynamics (e.g. stickiness) of dough. Another option is to perform online system identification of the dough dynamics parameters [1,2] or real2sim methods [3, 4]. In future work, we can also integrate our method with other works that perform low-level dough manipulation in the real world, such as recent work from Qi et al. [5].
>
> > Concern 3: Requires to specify the number of steps during tests.
>
> We vary the number of planning steps for the CRS task and show the results in the table below. We can see that our method is robust to the number of planning steps. As such, our method does not require knowing the exact number of the planning step. One can specify a max number of steps for all tasks and still use PASTA for planning. Here, we use receding horizon planning, as we did for the other 6-step planning task.
>
> | Number of planning steps             | 3 Step | 4 Step | 5 Step | 6 Step |
> |--------------------------------------|:--------:|:--------:|:--------:|:--------:|
> | Performance (normalized improvement) | 0.896  | 0.866  | 0.90   | 0.878  |
> | Success Rate                         | 5/5    | 4/5    | 5/5    | 4/5    |
>
> > Design choice 1: Why use the tool 'pointcloud' for Point Cloud Policy instead of the estimated pose and the type of the tool. How robust is cloud-based tool input regarding the tool occlusion caused by the robot arm/gripper and dough?
>
> Please see Table 5 of our Appendix, Section 4.1, where we compare using the tool point cloud as input to the policy (“PASTA (Ours)”) with using the tool pose as input to the policy (“Tool Concat Policy”). PASTA achieves a performance of 0.837 while the Tool Concat Policy only achieves a performance of 0.516. We speculate that this is because it is easier for the policy to reason about the interaction between the tool and the dough when they are both represented as point clouds.
>
> In the real world, we can recover the full tool pointcloud given the pose of the robot and a model of the tool. In this case, occlusion of the tool won’t be an issue. A similar policy is used in Qi et al. [5] and transferred to the real world.
>
> > Design choice 2: Why do you think decoupling subgoal and skill sequence for optimization is a great choice? Why not optimize them simultaneously?
>
> To clarify: we perform a joint, not sequential, optimization. In other words, we do not optimize the skill sequence first and then fix it while optimizing the subgoals; from a mathematical perspective, we optimize them jointly as an objective of: $\max_{subgoal, skill sequence}$. In practice, we implement this joint optimization by iterating over all skill sequences and optimizing the subgoals for each one, and then choosing the global max over all skill sequences. We can explore more efficient optimization algorithms, such as incremental planning and lazy planning [6] in future work. We have updated our limitations section of our paper based on this point.

---

> > ### Author Response · Authors · 2022-08-25
> > **Author response (Part 2 of 2)**
> >
> > > Design choice 3: Why is it better to use a reward 'predictor'? Can we convert the decode the latent codes and directly get the reward?
> > We learn the reward predictor because decoding each latent vector would greatly bottleneck the planning speed. Experiments on CutRearrange show that with our learned reward predictor, the planning takes 35s; on the other hand, if we decode the latent vectors and use the Chamfer Distance, even with a subsample point cloud of 200 points, the planning takes 37200s (around 10 hours), which is impractical to use.
> >
> > Moreover, using a reward predictor can also offer us the flexibility to incorporate more complex reward functions in the future.
> > We have added this discussion to the paper in Sec. 3.2 and to the Appendix in Sec. 1.5.
> >
> > > Issue on evaluation: heuristic controllers in the real world might conceal the sub-goal and action error, e.g. Fig 3, CutRearrange.
> >
> > We thank the reviewer for pointing this out. The alignment of the observation with the goal in Fig 3, CutRearrange Step 2 is just a coincidence. We have replaced the figure with a more representative example. To give a more comprehensive illustration of our planning result in the real world, we have included the generated subgoals for every evaluation rollout on our website: https://sites.google.com/view/pasta-plan (You can click on the navigation bar on the top to see the rollouts for each environment separately: [CutRearrange rollouts](https://sites.google.com/view/pasta-plan/cutrearrange-rollouts?authuser=0), [CRS rollouts](https://sites.google.com/view/pasta-plan/crs-rollouts), [CRS-Twice rollouts](https://sites.google.com/view/pasta-plan/crs-twice-rollouts)). These generated subgoals are a demonstration that our planner works in the real world.
> >
> > We also quantitatively computed the action error v.s. subgoal error for our real world trajectories. The results are shown below. From the results in the table, our planned goal is closer to the ground truth goal than the achieved goal, measured by the Earth Mover’s Distance (EMD), which shows that the controller does not compensate for the error of the planner.  We’ve also added this result to Sec. 4.4 in the Appendix.
> > |                                      |  CutRearrange |      CRS      |   CRS-Twice   |
> > |:------------------------------------:|:-------------:|:-------------:|:-------------:|
> > | EMD(planned goal, ground-truth goal) | 0.038 ± 0.004 | 0.027 ± 0.004 | 0.029 ± 0.002 |
> > | EMD(reached goal, ground-truth goal) | 0.056 ± 0.007 | 0.044 ± 0.006 | 0.054 ± 0.016 |
> >
> > > More details: Explicitly making a section describing the training process and the entire loss formulation
> >
> > We have included additional information about the training data and described our training procedures in detail in Sec 1.5 of the Appendix. We have also added the loss function for each module (the VAE, policy, feasibility predictor, and reward predictor) in Sec 1.5 of the Appendix.
> >
> > Please let us know if your concerns are addressed and please consider raising your score if you are satisfied. We are happy to answer any further questions you may have.
> >
> > ---
> > References
> >
> > [1] Yu, Wenhao, et al. "Preparing for the unknown: Learning a universal policy with online system identification." RSS 2017
> >
> > [2] Kumar, Ashish, et al. "Rma: Rapid motor adaptation for legged robots." RSS 2021
> >
> > [3] Ramos, Fabio, Rafael Carvalhaes Possas, and Dieter Fox. "Bayessim: adaptive domain randomization via probabilistic inference for robotics simulators." RSS 2019
> >
> > [4] Chebotar, Yevgen, et al. "Closing the sim-to-real loop: Adapting simulation randomization with real world experience." ICRA 2019
> >
> > [5] Qi, Carl, et al. "Learning Closed-Loop Dough Manipulation Using a Differentiable Reset Module." R-AL 2022
> >
> > [6] Garrett, Caelan, Tomás Lozano-Pérez, and Leslie Kaelbling. "Sample-based methods for factored task and motion planning." RSS 2017

---

### Official Review · Reviewer_ZcC8 · 2022-07-29

**Originality:** Good
**Technical Quality:** Very Good
**Clarity Of Presentation:** Very Good
**Impact:** 2

**Recommendation:**

Weak Accept: I recommend accepting the paper, but will not argue for my recommendation if the majority of other reviewers have a different opinion.

**Summary:**

This paper presents a learning pipeline that combines spatial and temporal representation to conduct long-horizon planning for deformable object manipulation.

**Issues:**

- Limitations could be elaborate on a bit more.
- The paper considers 3 skills, respectively push, cut, and spread, and the longest task requires applying 6 skills in sequence. This makes the possible combination of actions #3 * 2^5# = 96 variations. This is a rather small search space to find the correct sequence, the reviewer wonders if the authors could comment on how well the method scale to larger search space for task sequencing.

**Quality Of The Limitations Section:**

Additional details required

**Reviewer Expertise:**

3: The reviewer is fairly confident that the evaluation is correct

**Robotics Focus:**

Sufficient demonstration on hardware

**Strengths And Weaknesses:**

Strength:
The paper is very well written and easy to follow. The framework is extensively tested with simulation and real robot experiments. The website and appendix provide complementary information which helps the reviewer to better understand the proposed approach.

Weaknesses:

The reviewer found the Limitations paragraph unsatisfactory. The first limitation talks about the limited generalization capability of learned parameters to different tasks which are acceptable. The second limitation on the sim-to-real gaps seems to be more like a statement rather than pointing out the limitation.

Minor:
- P.4. Line 99, one of the K skills, K is not defined.
- P.4. Line 122-123 “a single-stage task with a single tool and can be just for a single object”, maybe it is better just “a single-stage task with a single tool for a single object”
- P.5. Alg 1  Decode subgoal from $U^h$ should be $U_h$?

**Summary Of Recommendation:**

A decent paper with extensive validation on simulation and real robot experiments.

---

> ### Author Response · Authors · 2022-08-25
> **Author response**
>
> Thank you for spending the time reviewing our paper and for providing constructive feedback. We have incorporated all your suggestions on our paper writing and uploaded a new version of our manuscript. The differences are marked in magenta. We respond to your questions below:
>
> >Limitations paragraph unsatisfactory
>
> We have uploaded a new version of the manuscript to address this concern. The limitation paragraph has been rewritten in which we describe the limitations of our method more clearly, including the limitations that hinder sim2real transfer.
>
> > the reviewer wonders if the authors could comment on how well the method scales to larger search space for task sequencing.
>
> PASTA has a computational complexity of $\mathcal{O}(K^H\cdot|\bf{I}|\cdot T_G)$, where $K$ is the number of skills, $H$ is the planning horizon, $|\bf{I}|$ is the number of attention structures and $T_G$ is the time it takes to solve each gradient-based optimization. As such, our planning method would not scale well if the length of the skill sequence becomes very large.
>
> Planning skill sequences with a large search space is a challenging problem by itself but much progress has been made by the task and motion planning community to obtain a plan skeleton [1,2,3]. For example, Caelan et al. [1] proposes to interleave searching the skill sequence with lower-level optimization or to have lazy placeholders for some skills. Danny et al. [3] proposes to predict skill sequences from visual observation. Recent works have also explored finding skill sequences using pre-trained language models [4, 5]. These methods can be incorporated into our framework in the future to scale our method to longer skill sequences. We have incorporated this discussion in the updated limitation section and in Section 5 of the appendix.
>
> Please let us know if your concerns are addressed and consider raising your score. We are happy to answer any further questions and improve the paper.
>
> ---
> References
>
> [1] Garrett, Caelan, Tomás Lozano-Pérez, and Leslie Kaelbling. "Sample-based methods for factored task and motion planning." RSS 2017.
>
> [2] Kim, Beomjoon, and Luke Shimanuki. "Learning value functions with relational state representations for guiding task-and-motion planning." Conference on Robot Learning. PMLR, 2020.
>
> [3] Driess, Danny, Jung-Su Ha, and Marc Toussaint. "Deep visual reasoning: Learning to predict action sequences for task and motion planning from an initial scene image." RSS 2020.
>
> [4] Ahn, Michael, et al. "Do as i can, not as i say: Grounding language in robotic affordances." arXiv preprint arXiv:2204.01691 (2022).
>
> [5] Huang, Wenlong, et al. "Language models as zero-shot planners: Extracting actionable knowledge for embodied agents." ICML 2022.

---

### Official Review · Reviewer_zTGC · 2022-07-30

**Originality:** Very Good
**Technical Quality:** Very Good
**Clarity Of Presentation:** Fair
**Impact:** 4

**Recommendation:**

Weak Accept: I recommend accepting the paper, but will not argue for my recommendation if the majority of other reviewers have a different opinion.

**Summary:**


This paper presents a planning framework for long-horizon manipulation over deformable objects. The approach authors present demonstrate good generalization ability to unseen number of objects and can incorporate quite a number of skills that involve deformable objects manipulation. The paper is important to the community in terms of scaling deformable object manipulation to long-horizon execution instead of always focusing on single-stroke, single-object manipulation.

**Issues:**

I hope most of my comments in the "weakness" part can be addressed.

**Quality Of The Limitations Section:**

Limitations are not well addressed

**Reviewer Expertise:**

5: The reviewer is absolutely certain that the evaluation is correct and very familiar with the relevant literature

**Robotics Focus:**

Sufficient demonstration on hardware

**Strengths And Weaknesses:**


Strengths:
- The paper presents an approach that shows quite an impressive plan over deformable object manipulation.  The experimental results and the videos are quite impressive. I think the authors are tackling important questions in terms of deformable object manipulation.
- The approach is designed in a way that it can learn skills in a simple simulator setting and use the skills in the unseen configurations which contain more objects.

Weakness:
I have quite a lot of comments on the paper writing. I think the paper writing needs improvements in terms of precise wording and related work citations:
- Line 4: I suggest authors can write more explicitly why "full-state" or "short-horizon" assumptions prevent the use of deformable objects
- Line 8: are authors using any other 3D observations? If not, "such as" is not precise here.
- Line 22: Are the skills (temporal abstractions) for deciding which tools to use? or the abstractions of execution with skills? Make discrete decisions of which tool to use also sounds like a single-step decision.
- Line 27: One issue I have is, why "atoms"? I think the community typically compares state abstractions with physical states or raw sensory representations. I don't think anyone does "infinitesimally" timesteps. The logic sounds ok but quite trivial. I think the authors can do much better in describing this part.
- Line 28-35: I found this paragraph overclaiming. A lot of works use temporal abstractions in robot research and spatial abstractions. I think the authors are considering the context of deformable objects. Please change "complex objects" to "deformable objects" to make this paragraph sound specific and precise.
- Line 40: What is "nearly infinite" degrees of freedom? Authors should provide a citation that strictly defines such an ambiguous term. Otherwise, I think “infinite” is sufficient
- Line 58: Task and Motion Planning doesn't construct the models --- they assume/use the models given.
- Line 72-78: missing citation -- such as regression planning network[1], which is grounds in visual observation but generalizes to long-horizon tasks.
- Line 120: I am not very convinced by the term. It seems to describe a single abstraction that is both spatial and temporal. But from my understanding of the paper, spatial abstraction is on the 3D representation, and temporal abstraction is about the tool-use skills. I don't think this is an appropriate name for abstractions in the paper.
- Line 168: Reward is quite an RL-oriented terminology, while the work focuses more on the planning work. I suggest using the term "cost" to better align with the planning literature. Also, that would align better with how pseudo code presents the approach.
- And I think the specification of goals is not ideal. The goal is given in terms of point clouds, which means that the execution of the task is required before planning a sequence for such an execution. While the goal specification is not the main point of this paper, it hinders the proposed approach's significance. Specifying goals simply could make the paper to be more significant. I think the authors should also discuss this limitation in Section 5. In my opinion, the unsupervised learning part is quite minor compared to the limitation of goal specification.


[1] Xu et al. Regression Planning Networks



**Summary Of Recommendation:**

Overall, I would like to give weak accept to this submission based on its impressive results and the hard problem the authors try to solve. However, I think the paper presentation needs a boost in quality in terms of precise wording.

---

> ### Author Response · Authors · 2022-08-25
> **Author response**
>
> **Comment:**
>
> Thank you for your valuable feedback. We have incorporated all your suggestions on our paper writing and uploaded a new version of our manuscript. The differences are marked in magenta. Below, we briefly respond to your comments:
>
> > missing citation -- such as regression planning network, which is grounded in visual observation but generalizes to long-horizon tasks.
>
> Thank you for pointing us to this related paper. We have cited this paper in our updated paper.
>
> Regression planning network learns a planner grounded on object-centric visual observation and can solve long-horizon tasks. However, its planner takes a symbolic goal (e.g. On(pot, stove)∧¬IsOpen(fridge)) and plans in a predefined symbolic domain to output symbolic subgoals. Such symbolic planning is difficult to extend to deformable objects such as dough. In contrast, our method does not require defining a symbolic planning domain.
>
> >I am not very convinced by the term (Neural Spatial-temporal Abstraction). It seems to describe a single abstraction that is both spatial and temporal.
>
> Thank you for the feedback.  Following your suggestion, we have changed all “spatial-temporal abstraction” in our paper to “spatial and temporal abstraction”.
>
> We also want to clarify that our method is not a simple combination of the two abstractions. Instead, the two abstractions are deeply integrated. Our temporal abstraction involves learning a set feasibility predictor and set reward predictor (now called “set cost predictor”). Their architectures are designed to be permutation invariant, which is due to the use of the latent set representation, i.e. our spatial abstraction. Second, we train the feasibility predictor to model the temporal abstraction of skills and it only takes in a subset of all the entities as input, enabling generalization to more entities. This compositional generalization ability of the feasibility predictor is also enabled by our spatial abstraction
>
> >What is "nearly infinite" degrees of freedom?
>
> In the most granular level, all objects can be represented by the atoms that consist of them. While this number is huge, it is bounded. As such, we use the word “nearly infinite” to be more precise.
>
> >And I think the specification of goals is not ideal.
>
> To clarify, in our experiments, specifying the goal does not require execution of the planner. Instead, we specify goals as 3D point clouds of different geometric shapes such as boxes and spheres at specific locations.  As an example, with CRS (Cut, rearrange, spread), the piece that needs to be spread is specified as a thin and long box at the target location. We have included the 3D view of the goal in a zip attached to this comment, and we also have added this clarification to the paper in Sec. 4.1.
>
> In our method, the goal is specified as a point cloud. Being able to specify the goal using point clouds and geometric shapes is very useful for tasks requiring fine granularity, which is especially true for deformable object manipulation. Furthermore, the goal-conditioned setting is also widely used in other studies [2, 3]
>
> Another way of specifying the goal can be using natural language [1]. Natural language is useful for  describing high-level goals but not as useful for specifying fine and detailed targets. We have added a discussion of goal specification to the limitations discussion in Section 5 of the appendix.
>
> Please let us know if your concerns are addressed and please consider raising your score if you are satisfied. We are happy to answer any further questions you may have.
>
> ---
> References
>
> [1] Cui, Yuchen, et al. "Can Foundation Models Perform Zero-Shot Task Specification For Robot Manipulation?." Learning for Dynamics and Control Conference. PMLR, 2022.
>
> [2] Andrychowicz, Marcin, et al. "Hindsight experience replay." Advances in neural information processing systems 30 (2017).
>
> [3] Nair, Ashvin V., et al. "Visual reinforcement learning with imagined goals." Advances in neural information processing systems 31 (2018).
>
>
> **Zip File:**
>
> /attachment/8aa9c4ac8a7dad6aed11d39f21d31de42107e203.zip

---

### Official Review · Reviewer_E6Aq · 2022-07-31

**Originality:** Very Good
**Technical Quality:** Very Good
**Clarity Of Presentation:** Very Good
**Impact:** 4

**Recommendation:**

Strong Accept: I recommend accepting the paper and will argue for my recommendation even if other reviewers hold a different opinion.

**Summary:**

This paper introduces a method that leverages spatial abstraction and temporal abstraction to facilitate planning for long-horizon deformable object manipulation. Temporal abstraction is achieved using primitive skills and spatial abstraction is achieved by independently representing each object in a scene. More specifically, the proposed model consists of 1) a Point Cloud VAE that encodes the point cloud of each object into a latent vector and can be used to generate imagined subgoals, 2) a set point cloud policy for each skill that operates on a set of object point clouds, 3) a set feasibility predictor that operates on latent codes of objects to predict skill feasibility, and 4) a set reward predictor that predicts distance to the goal. A planning algorithm uses the learned components to predict a sequence of primitive skills and the latent subgoals for the skills. The latent subgoals are decoded with the VAE decoder and used to condition the set point cloud policy.


**Issues:**

Besides the questions I raised above, I have the following comments:
1. In Line 91, $T$ is used to denote the number of actions between the initial and goal. $T$ is used later to represent the skill horizon. Since multiple skills are executed between the initial step and the goal step, maybe a different symbol should be used.
2. In Line 92, $i$ is used to index time but it is later used to index objects.
3. Line 117 is confusing since it states that the point cloud of each object is encoded in a set of latent vectors.
4. Line 130 states that the set point cloud policy takes in the observed point cloud $P_{obs}$. Doesn’t the policy network take in a set of object point clouds {$P_1,...,P_C$} instead? I recommend the authors more clearly differentiate point cloud sets and the point cloud of the whole scene in the method section.
5. Line 175, why is the MLP trained to predict Chamfer distance between latent vectors?
6. Why did PASTA achieve better performance on CutRearrangeSpread(CRS) than CutRearrange? Does this have to do with the different training for the two tasks?
7. Looking at Step 3 of CRS-Twice in Figure 3, the predicted subgoal does not match the actual point cloud of the left object but this task is still completed successfully. Can the authors provide more insight into the robustness of the algorithm to errors in subgoal prediction and whether the policy networks are using the predicted subgoals?


**Quality Of The Limitations Section:**

Additional details required

**Reviewer Expertise:**

4: The reviewer is confident but not absolutely certain that the evaluation is correct

**Robotics Focus:**

Sufficient demonstration on hardware

**Strengths And Weaknesses:**

Strengths:
1. Compared to the most-related prior work [7], this paper introduces set-based versions of the skill policy, feasibility predictor, and reward predictor using clever designs of the networks, objective functions, and planning algorithm.
2. A nice property of the planning algorithm is that it can work with skills that change the number of objects in a scene.
3. The attention mask is used to exclude irrelevant parts of the scene. The importance of this design is validated in Table 4 in the appendix.
4. The results presented in Table 1 are very strong. The Flat3D and DiffSkill-Image baselines help to demonstrate the importance of spatial abstraction, which is the main contribution of this paper.
5. It’s great to see that this method also generalizes well to the real robot using heuristic controllers for the skills.
6. The discussion and ablation study of the feasibility predictor also provide a lot of insights on how to ensure its gradient can successfully guide the search for correct subgoals.

Weakness:
1. It would be great to see a discussion of the complexity of the algorithm in the main text. Since the planning algorithm performs nested optimization at 3 different levels. I am curious if the algorithm can easily scale to tasks that require chaining more than 6 skills (e.g., CRS-Third, CRS-Fourth, and etc) and more than 3 objects. Will the algorithm work if there are distractor objects in the scene?
2. It would be interesting to test the generalization capability of the proposed method with more tasks. Can the method be evaluated on different combinations of the primitive skills (i.e., Lift, Spread, Gather, Transport, Cut, Rearrange).


**Summary Of Recommendation:**

This paper combines spatial and temporal abstraction to achieve long-horizon deformable manipulation. The proposed planning framework and neural network modules can deal with variable numbers of objects and compose primitive skills to manipulate more objects in test time. The experiment shows that this object-centric scene representation is more effective than existing methods that do not use spatial abstraction. The real robot experiment also validates the proposed method.

---

> ### Author Response · Authors · 2022-08-25
> **Author response (Part 1 of 2)**
>
> Thank you for your valuable feedback. We have addressed all your comments on our paper and marked the revised texts in magenta. Below, we make further clarifications to these comments:
> >It would be great to see a discussion of the complexity of the algorithm in the main text. Since the planning algorithm performs nested optimization at 3 different levels. I am curious if the algorithm can easily scale to tasks that require chaining more than 6 skills (e.g., CRS-Third, CRS-Fourth, and etc) and more than 3 objects.
>
> We have discussed the complexity of our algorithm in the revised paper (Around line 250 of the main text). PASTA has a computational complexity of $\mathcal{O}(K^H\cdot|\bf{I}|\cdot T_G),$ where $K$ is the number of skills, $H$ is the planning horizon, $|\bf{I}|$ is the number of attention structures and $T_G$ is the time it takes to solve each gradient-based optimization. As such, scaling to longer sequences can take a very long time, but choosing from more skills for a short sequence can be done within a shorter time. Our method can scale to more objects as each skill only needs to focus on a few objects at a time.
>
> Planning skill sequences with a large search space is a challenging problem by itself but much progress has been made by the task and motion planning community to obtain a plan skeleton [1,2,3]. For example, Caelan et al. [1] proposes to interleave searching the skill sequence with lower-level optimization or to have lazy placeholders for some skills. Danny et al. [3] proposes to predict skill sequences from visual observation. Recent works have also explored finding skill sequences using pre-trained language models [4, 5]. These methods can be incorporated into our framework in the future to scale our method to longer skill sequences.  We have incorporated this discussion in the updated limitations section as well as in section 5 of the Appendix.
> >Will it work if there are distractor objects in the scene?
>
> We conducted additional experiments by adding 2 distractor objects in CRS (which makes the scene have 4 objects in total). We observe that PASTA retains a normalized performance of 0.879 and 100% success rate using the same amount of samples to plan. Our planner is able to ignore the distractors using our attention structure at every step to only attend to the relevant components in the scene. We also added an example trajectory with distractor dough pieces to our website under [*CRS with distractors*](https://sites.google.com/view/pasta-plan/home#h.qhkvbhxdpdtl).
> >It would be interesting to test the generalization capability of the proposed method with more tasks. Can the method be evaluated on different combinations of the primitive skills (i.e., Lift, Spread, Gather, Transport, Cut, Rearrange).
>
> This would be interesting.  We can explore such combinations in future work, though we do not see any obvious reason why this would not work at the moment.
> >Why did PASTA achieve better performance on CutRearrangeSpread (CRS) than CutRearrange? Does this have to do with the different training for the two tasks?
>
> We use the same training procedure for all tasks. Here, CutRearrange achieves a success of 4/5 while CRS achieves a success of 5/5. The one “failure” trajectory for CutRearrange has a normalized performance of 0.756, which is fairly close to the threshold of success (0.8) and still above the mean of all the baselines. Therefore, we think that this is due to slight imperfections of the model in CutRearrange and does not result from the training procedure.

---

> > ### Author Response · Authors · 2022-08-25
> > **Author response (Part 2 of 2)**
> >
> > >Predicted subgoal does not match the actual point cloud but the task is still completed successfully, e.g. Fig 3, CRS-Twice.
> >
> > We thank the reviewer for pointing this out. The alignment of the observation with the goal in Fig 3, CRS-Twice Step 3 is just a coincidence. We have replaced the figure with a more representative example. To give a more comprehensive illustration of our planning result in the real world, we have included the generated subgoals for every evaluation rollout on our website: https://sites.google.com/view/pasta-plan (You can click on the navigation bar on the top to see the rollouts for each environment separately: [CutRearrange rollouts](https://sites.google.com/view/pasta-plan/cutrearrange-rollouts?authuser=0), [CRS rollouts](https://sites.google.com/view/pasta-plan/crs-rollouts), [CRS-Twice rollouts](https://sites.google.com/view/pasta-plan/crs-twice-rollouts)). These generated subgoals are a demonstration that our planner works in the real world.
> >
> > We also quantitatively computed the action error v.s. subgoal error for our real world trajectories. The results are shown below. From the results in the table, our planned goal is closer to the ground truth goal than the achieved goal, measured by the Earth Mover’s Distance (EMD), which shows that the controller does not compensate for the error of the planner.  We’ve also added this result to Sec. 4.4 in the Appendix.
> > |                                      |  CutRearrange |      CRS      |   CRS-Twice   |
> > |:------------------------------------:|:-------------:|:-------------:|:-------------:|
> > | EMD(planned goal, ground-truth goal) | 0.038 ± 0.004 | 0.027 ± 0.004 | 0.029 ± 0.002 |
> > | EMD(reached goal, ground-truth goal) | 0.056 ± 0.007 | 0.044 ± 0.006 | 0.054 ± 0.016 |
> >
> > ---
> > References:
> >
> > [1] Garrett, Caelan, Tomás Lozano-Pérez, and Leslie Kaelbling. "Sample-based methods for factored task and motion planning." RSS 2017
> >
> > [2] Kim, Beomjoon, and Luke Shimanuki. "Learning value functions with relational state representations for guiding task-and-motion planning." Conference on Robot Learning. PMLR, 2020.
> >
> > [3] Driess, Danny, Jung-Su Ha, and Marc Toussaint. "Deep visual reasoning: Learning to predict action sequences for task and motion planning from an initial scene image." RSS 2020.
> >
> > [4] Ahn, Michael, et al. "Do as i can, not as i say: Grounding language in robotic affordances." arXiv preprint arXiv:2204.01691 (2022).
> >
> > [5] Huang, Wenlong, et al. "Language models as zero-shot planners: Extracting actionable knowledge for embodied agents." ICML 2022.

---

### Author Response · Authors · 2022-08-25
**Revised paper, revised appendix and summary of the changes**

**Comment:**

We thank the area chair and all the reviewers for their valuable feedback. **The revised paper and appendix are attached in this thread.** In addition to the response to specific reviewers, here we summarize the added experiments and revision to the paper.
First, we have incorporated all reviewer’s comments about writing, including:
* more discussion on our limitations
* computation complexity of our algorithm
* modification of the terminology
* more clarification of the method

These changes are marked in magenta in our revised paper and the Appendix.

We have also modified Fig. 3 to use a more representative example and added generated plans for every trajectory on our website ([CutRearrange rollouts](https://sites.google.com/view/pasta-plan/cutrearrange-rollouts?authuser=0), [CRS rollouts](https://sites.google.com/view/pasta-plan/crs-rollouts), [CRS-Twice rollouts](https://sites.google.com/view/pasta-plan/crs-twice-rollouts)) to give a more comprehensive illustration of our real world experiments.


**Zip File:**

/attachment/0277bbf0c6a791ad56c915fa92ebac934e4726bc.zip

---

### Meta-Review · Area_Chair_JLzu · 2022-08-15

**Recommendation:** Accept (Poster)
**Confidence:** 4

**Metareview:**

This paper presents a planning approach that learns spatial/temporal abstractions of a set of skills to cut, spread and push deformable objects from demonstration.

The quality and originality of the paper are very good. Also, the significance of the work is high with the potential to have a major impact on the robotics learning community.

Overall the approach is sound and the results are impressive, most of the clarifications/minor issues from the reviewers were addressed successfully.

The supplementary material is to be commended.

**Best Paper Nomination:**

No